# Symmetry-breaking host–guest assembly in a hydrogen-bonded supramolecular system

Shinnosuke Horiuchi [1,2,3] ✉, Takumi Yamaguchi [4], Jacopo Tessarolo [2], Hirotaka Tanaka[1], Eri Sakuda[1,2], Yasuhiro Arikawa [1], Eric Meggers [5], Guido H. Clever [2] ✉ & Keisuke Umakoshi [1] ✉

Bio-inspired self-assembly is invaluable to create well-defined giant structures from small molecular units. Owing to a large entropy loss in the self-assembly process, highly symmetric structures are typically obtained as thermodynamic products while formation of low symmetric assemblies is still challenging. In this study, we report the symmetry-breaking self-assembly of a defined $C_1$-symmetric supramolecular structure from an $O_h$-symmetric hydrogen-bonded resorcin[4]arene capsule and $C_2$-symmetric cationic bis-cyclometalated Ir complexes, carrying sterically demanding tertiary butyl ($^t$Bu) groups, on the basis of synergistic effects of weak binding forces. The flexible capsule framework shows a large structural change upon guest binding to form a distorted resorcin[4]arene hexameric capsule, providing an asymmetric cavity. Location of the chiral guest inside the anisotropic environment leads to modulation of its Electric Dipole (ED) and Magnetic Dipole (MD) transition moments in the excited state, causing an increased emission quantum yield, longer emission lifetime, and enhancement of the dissymmetry factor ($g_{lum}$) in the circularly polarized luminescence.

Multi-component self-assembly via reversible bond formation and non-covalent interactions typically affords highly symmetric structures, as commonly seen in nature[1–3]. Over the last four decades, extensive studies of coordination driven self-assembly have demonstrated methods to form a large variety of spheres, octahedra, cubes, tetrahedra and similar regularly shaped architectures consisting of rigid organic ligands and metal ions[4–6]. Recently, to increase complexity and functionality in such coordination-based nano-systems, several strategies that make use of mixtures of ligands have been developed[7], including coordination sphere engineering[8–12], shape-complementary assembly[13–16], the use of non-symmetric ligands[17–19], and backbone-centered steric hindrance[20]. However, multi-component self-assembly forming a $C_1$-symmetric structure is still challenging owing to a large entropic disadvantage[21,22].

The three dimensional structures of biomolecules constructed by weak associating forces, such as hydrogen-bonds, cation-π, anion-π, and CH-π interactions, which are usually considerably weaker than metal–ligand coordination bonds, can often exhibit large structural changes promoted via guest recognition at their binding sites, called induced-fit and allosteric mechanisms[23,24]. In many cases, rigid organic substrates will induce reorganization of the flexible binding pocket of an enzyme to increase guest affinity. In a similar way, the cooperative action of guest recognition and structural adaptivity within self-assembled 3D structures constructed by weak associating forces can endow a supramolecular system with additional complexity, providing

[1]Division of Chemistry and Materials Science, Graduate School of Engineering, Nagasaki University, 1-14 Bunkyo-machi, Nagasaki 852-8521, Japan. [2]Department of Chemistry and Chemical Biology, TU Dortmund University, Otto-Hahn-Straße 6, 44227 Dortmund, Germany. [3]Department of Basic Science, Graduate School of Arts and Sciences, The University of Tokyo, 3-8-1 Komaba, Meguro-ku, Tokyo 153–8902, Japan. [4]School of Materials Science, Japan Advanced Institute of Science and Technology, 1-1 Asahidai, Nomi 923–1292, Japan. [5]Fachbereich Chemie, Philipps-Universität Marburg, Hans-Meerwein-Straße 4, 35043 Marburg, Germany. ✉e-mail: shoriuchi@g.ecc.u-tokyo.ac.jp; guido.clever@tu-dortmund.de; kumks@nagasaki-u.ac.jp

a chance to afford an unusual low-symmetry structure as a thermodynamic product.

Resorcin[4]arene **1**, one of the most classical compounds applied in supramolecular chemistry, is known to assemble into a highly symmetrical hexameric capsule in apolar solvents ((**1**)$_6$, $O_h$ symmetry), comprising eight water molecules which are incorporated via a multitude of hydrogen-bonds (Fig. 1a)[25]. The resorcin[4]arene hexamer (**1**)$_6$ features an octahedral cavity (1300 Å$^3$) which was shown to exhibit remarkable catalytic activity for the conversion of small organic substrates with unusual product selectivity and rate acceleration, similar to enzymatic transformations[26–28]. The electron-rich aromatic walls, allowing cation-π interactions, and trapped water molecules which can act as mild Brønsted acids (p$K_a$ = 5.5 ~ 6) within the capsule framework were identified as key factors in this system. Our group has previously utilized this hydrogen-bonded hexameric capsule for the encapsulation of luminescent coordination complex salts, forming supramolecular complexes insulated from bulk solvent. As a result, encapsulation-induced emission enhancement (EIEE) behavior was observed for the guest complex[29–32].

Encapsulation of tertiary ammonium cations, which are typical guests for the hexameric capsule, has been mainly studied in solution and in the solid state[33]. Little is known about the gas phase structures of such host–guest complexes as they easily show fragmentation under usual mass spectrometry measurement conditions such as

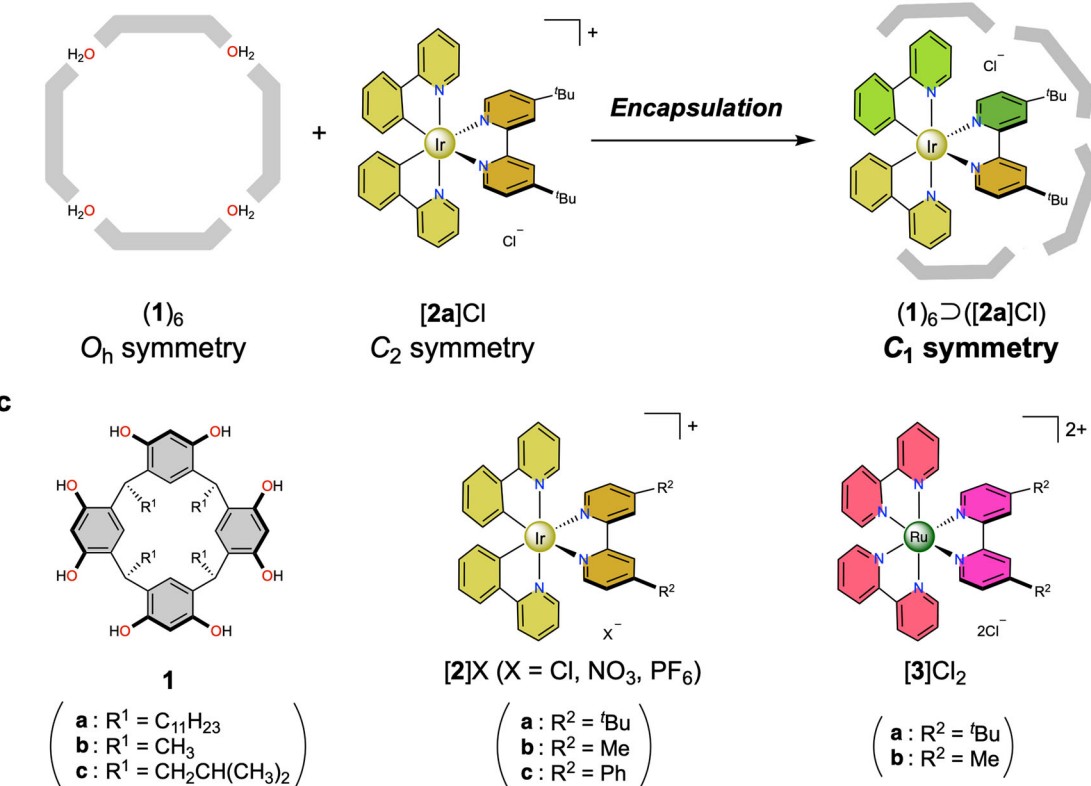

**Fig. 1 | Self-assembly of hydrogen-bonding resorcin[4]arene host. a** Formation of a hexameric capsule with eight water molecules. **b** Symmetry-breaking assembly via host–guest complexation. **c** Compounds used in this study.

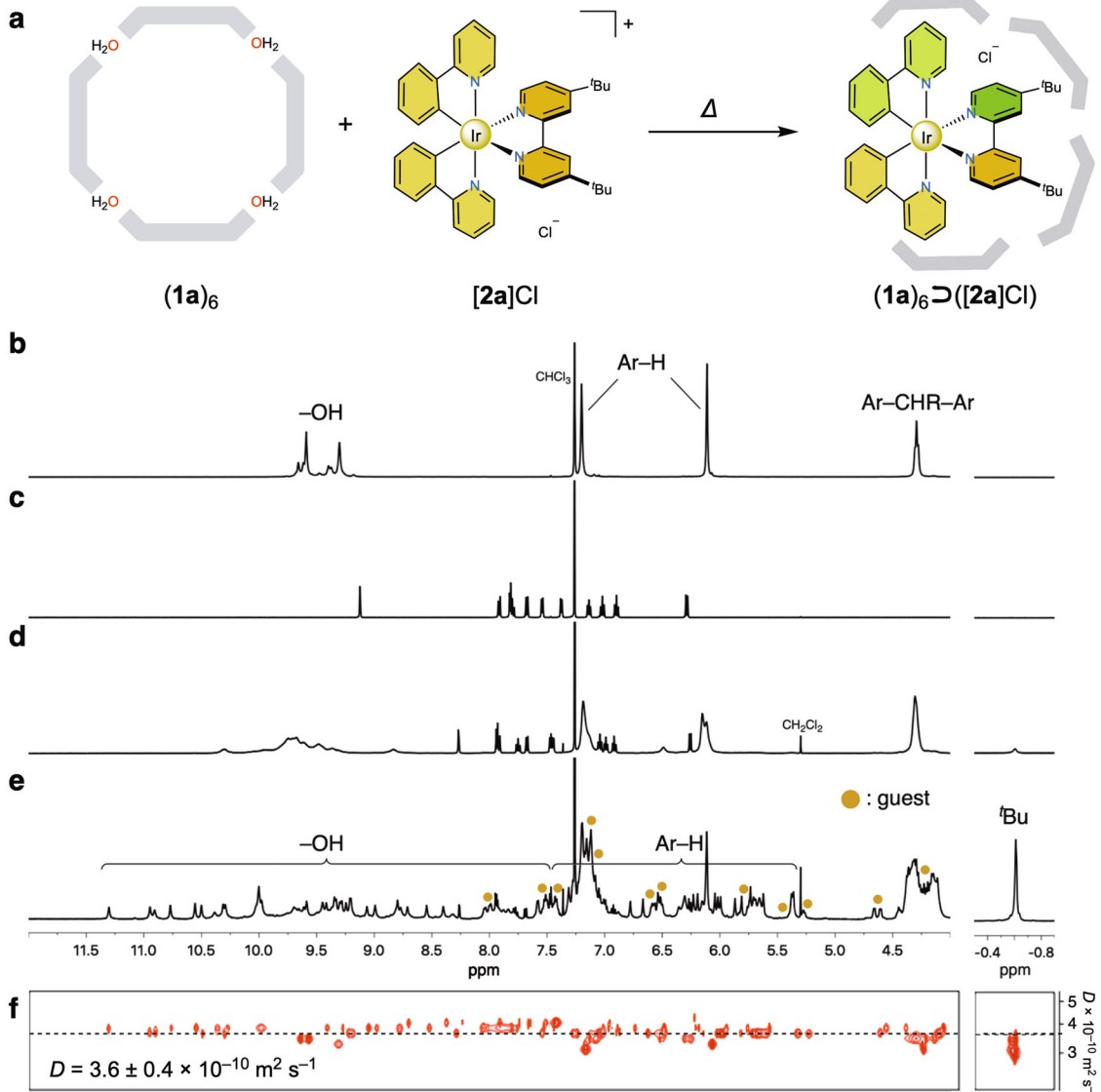

**Fig. 2 | Symmetry-breaking assembly consisting of resorcin[4]arene and cationic Ir complex. a** Schematic representation of the assembly. $^1$H NMR spectra (500 MHz, CDCl$_3$, RT) of **b** resorcin[4]arene hexameric capsule (**1a**)$_6$, **c** Ir complex [**2a**]Cl, **d** a mixture of (**1a**)$_6$ and [**2a**]Cl, **e** the mixture after heating at 50 °C for 1 h, and **f** partial $^1$H DOSY spectrum of the host−guest complex showing the diffusion coefficient ($D$) around $3.6 \pm 0.4 \times 10^{-10}$ m$^2$ s$^{-1}$. The signals marked with beige filled circles indicate the encapsulated Ir complex.

electrospray ionization[34]. In contrast, cationic coordination complexes possess larger and more rigid structures as compared to tertiary ammonium cations, which can serve as templates to strongly stabilize discrete resorcin[4]arene oligomers even in the gas-phase[34]. This observation can be explained by a snug fit of the guest's large surface area, giving rise to significant stabilization via cation-π and CH−π interactions.

Herein we report the symmetry-breaking behavior of supramolecular complexes forming from a hexameric resorcin[4]arene capsule and luminescent coordination complexes via molecular recognition (Fig. 1b). The coordination complex, carrying sterically demanding substituents, causes a significant structural perturbation of the hydrogen-bonded capsule, away from its typical and thermodynamically most stable solution structure with $O_h$ symmetry, to a defined low-symmetry assembly stabilized by noncovalent host-guest interactions and multiple hydrogen-bond networks. Hence, the $C_2$-symmetric coordination complex combined with the $O_h$-symmetric hydrogen-bonded resorcin[4]arene capsule leads to a symmetry-breaking phenomenon to afford a $C_1$-symmetric supramolecular

complex. Notably, this leads to a rare encapsulation phenomenon, EIEE, for the guest. Furthermore, the symmetry-breaking effect desymmetrizes ED and MD transition moments in the radiative process (emission) of the chiral-at-metal coordination compound within the resorcin[4]arene oligomer, resulting in an increase of the dissymmetry factor $g_{lum}$ in the complex's circularly polarized luminescence (CPL) property.

## Results

### Encapsulation of Ir complexes

The host−guest complex (**1**)$_6$⊃([**2**]Cl) was obtained via the following process regarding the self-assembly of (**1**)$_6$ and encapsulation of [**2**]Cl (Supplementary Fig. 2). First, simple mixing of CDCl$_3$ solutions of (**1a**)$_6$ and [**2a**]Cl in a 1: 1 ratio at room temperature showed upfield shifts of aromatic protons in the $^1$H NMR spectrum of [**2a**]$^+$, as compared with that of free [**2a**]Cl in CDCl$_3$ (Fig. 2b−d). The doublet signal at 9.12 ppm, assigned to the 3,3'-protons on the 2,2'-bipyridine moiety in [**2a**]Cl, is sensitive to the presence of a chloride ion owing to a strong chelating hydrogen-bond formed with the anion[35]. The resorcin[4]arene capsule

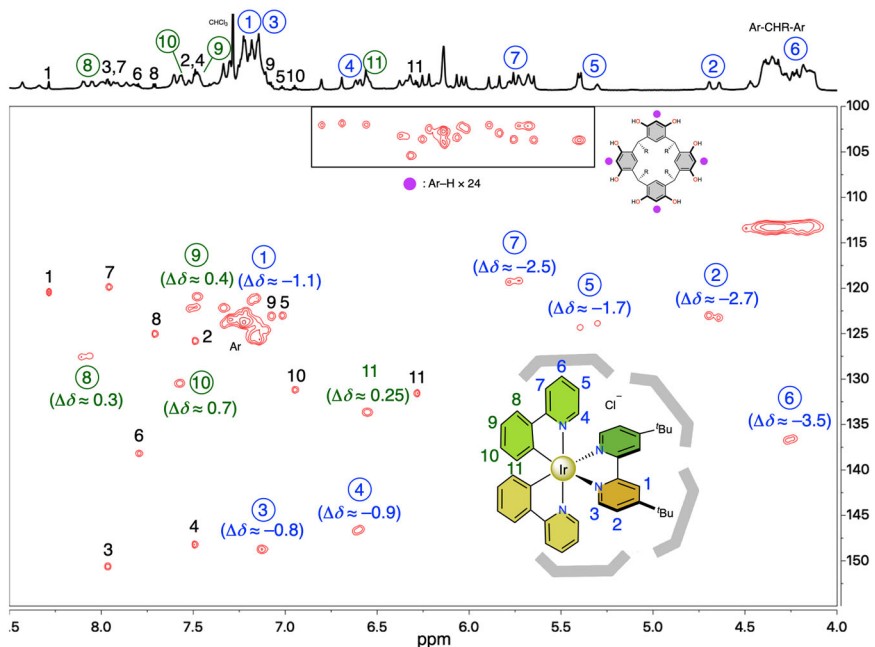

**Fig. 3 | $^1$H–$^{13}$C heteronuclear single quantum correlation (HSQC) spectrum ($^1$H: 800 MHz, CDCl$_3$, 298 K) of the mixture of [2a]Cl and (1a)$_6$⊃([2a]Cl).** The cross peaks marked with black and encircled numbers are assigned to the protons of free [2a]$^+$ and entrapped [2a]$^+$, respectively. The upfield (<0) and downfield shifts (>0) in the $^1$H NMR spectrum caused by the host–guest formation are shown as Δδ in parentheses.

initially expels the chloride ion from the complex cation because of the stronger hydrogen-bonding nature of phenolic protons in the hexameric capsule (1a)$_6$[31], leading to a significant upfield shift of the doublet peak in [2a]$^+$ after mixing the solution of (1a)$_6$ and [2a]Cl (Fig. 2d). During heating the mixture at 50 °C, partial dissociation of 1a from the assembly is likely to form a portal in the hydrogen-bonded hexameric capsule and encapsulation of [2a]$^+$ within the resorcin[4]arene capsule proceeds to afford the host–guest complex (1a)$_6$⊃([2a]Cl)[31]. The $^1$H NMR spectrum recorded after heating for 1 h clearly showed that a singlet peak of $^t$Bu substituents in [2a]$^+$ appeared at −0.61 ppm owing to the shielding effect from the host, suggesting that the Ir complex was trapped inside the resorcin[4]arene capsule (Fig. 2e). Very interestingly, the host–guest complex (1a)$_6$⊃([2a]Cl) showed severe signal splitting in the $^1$H NMR spectrum. This is probably caused by a strong template effect from [2a]$^+$ which can effectively suppress some molecular dynamics, such as self-assembly and encapsulation equilibria in (1a)$_6$⊃([2a]Cl), rapid rotation of [2a]$^+$ within the resorcin[4]arene capsule, and hydrogen-bond recombination of the capsule on the NMR time scale. A variable temperature (VT) NMR study revealed that the molecular dynamics were still slow even at elevated temperature on the NMR time scale (Supplementary Fig. 5). Diffusion-ordered NMR spectroscopy (DOSY) showed a single band at diffusion coefficient $D_{(host-guest)}$=3.6 ± 0.4 × 10$^{-10}$ m$^2$ s$^{-1}$ and significant reduction with respect to the diffusion coefficient of free [2a]$^+$ ($D_{(free\ Ir)}$=7.0 ± 0.5 × 10$^{-10}$ m$^2$ s$^{-1}$, $D_{(host-guest)}$/$D_{(free\ Ir)}$ ≈ 0.50), suggesting that the solution mainly contained a single supramolecular species and the Ir complex was strongly trapped within the resorcin[4]arene capsule via multiple cation-π interactions and ion-pair formation within the cavity (Fig. 2f and Supplementary Fig. 6).

## Symmetry analysis of the host-guest complex

The molecular symmetry of the resorcin[4]arene capsule drastically reduced from $O_h$ to $C_1$ symmetry by encapsulation of the large guest [2a]$^+$, judging from the severe peak splitting and the single diffusion coefficient of the host-guest complex in the $^1$H DOSY spectrum. The symmetry lowering of self-assembled molecular hosts has typically been observed in host-guest systems using rigid coordination-driven

metallocages, when guest molecules are tightly bound and the motions of guests are restricted within the cages[36–38]. Thus, this is indeed a rare NMR observation for the desymmetrization of an otherwise rather flexible resorcin[4]arene hydrogen-bonded system[39].

For symmetry analysis of the host–guest complex (1a)$_6$⊃([2a]Cl), $^1$H–$^{13}$C heteronuclear single quantum correlation (HSQC) spectroscopic analysis was utilized for the assignments of the NMR signals in this host–guest system (Fig. 3). The HSQC spectrum clearly discriminates the cross peaks derived from the host and the guest complex, clarifying the structural aspects of the assembly. The aromatic proton of free 1a at 6.2 ppm splits into 24 signals observed at around 5 to 7 ppm in the $^1$H NMR spectra (100 to 110 ppm for the corresponding carbon atoms in the $^{13}$C NMR spectra), indicating that all resorcin[4]arene units are fixed in non-equivalent environments around the guest on the NMR time scale (Supplementary Fig. 11).

Signals 1–7 assigned to the pyridine protons of [2a]$^+$ are considerably shifted to the upfield region (Δδ = −0.8 ~ −3.5 ppm) and, in contrast, protons 8–11 at the phenyl moieties show a downfield shift (Δδ = 0.25 ~ 0.7 ppm) on host–guest formation. As guest signals typically show upfield shifts upon encapsulation, owing to the shielding effect from the hosts, this HSQC spectrum suggests that the pyridine moieties of entrapped [2a]$^+$ were located within the hexameric capsule, while the phenyl moieties were not covered by the capsule. More interestingly, protons 1, 2 and 4–8 are clearly split into two peaks, suggesting that two ppy ligands and two pyridine moieties in the bpy ligand in [2a]$^+$ are located in an asymmetric environment generated by the distorted hexameric capsule to lower the molecular symmetry of the host–guest complex on the NMR time scale. Therefore, this HSQC result clearly indicates that the host–guest complexation of hexameric capsule (1)$_6$ with $O_h$ symmetry and [2a]Cl with $C_2$ symmetry leads symmetry-breaking to afford a $C_1$-symmetric compound as the thermodynamic product.

Coordination-driven supramolecular architectures were previously shown to undergo dynamic transformations via guest binding, as seen in induced-fit molecular recognition mechanisms[23, 24]. Such biomimetic functions belong to important challenges in supramolecular chemistry[40, 41]. Although individual association forces in the

system studied herein are weaker and less directional than typical coordination bonds, our results show that the cooperation of multiple weak interactions can control the structure and symmetry of the final product, in particular, achieving the formation of an unusual $C_1$-symmetric assembly.

### Solution structure of the distorted hexameric capsule

To gain further structural information about the symmetry-breaking assembly, we carried out several 2D NMR experiments for determining the relative orientation between the host and the guest molecules. A VT-HSQC measurement at 50 °C showed significant broadening of the peaks of the resorcin[4]arene molecules around 5.7 and 6.3 ppm, while other peaks were still sharp and slightly shifted (Supplementary Fig. 12). This suggests that two resorcin[4]arene units in the distorted hexameric capsule are relatively flexible, as compared with the other four resorcin[4]arene units. Furthermore, the Rotating-frame Overhauser Effect Spectroscopy (ROESY) spectrum showed that the flexible resorcin[4]arene units correlate to guest protons 8 to 11. Therefore, it was concluded that the flexibility of the host originates from weaker interactions of the host's electron-rich aromatic moieties with the guest.

A $^1$H–$^1$H Nuclear Overhauser Effect Spectroscopy (NOESY) experiment revealed that the sharp signal at −0.61 ppm in the $^1$H NMR spectrum, assigned to the $^t$Bu groups in [2a]$^+$, gives eight cross-peaks with rigid 1a (Supplementary Fig. 8). It suggests that two resorcin[4] arene units cover the $^t$Bu groups and fix their orientation owing to effective CH−π interactions. The signal at 5.4 ppm of 1a shows two cross-peaks with guest protons 3 and 8 in the ROESY spectrum, implying that the pyridine moieties in the guest are recognized by the concave cavities of the resorcin[4]arene molecules, probably because the electron-rich aromatic walls of the host effectively capture the electron-deficient pyridyl units (Supplementary Fig. 9).

We propose that these different binding modes concertedly serve to form an asymmetric cavity in the distorted hexameric capsule. A partial structure of such a binding motif could be visualized by a single crystal X-ray analysis of Ru complex analogue [3a]Cl$_2$ with 1c (Supplementary Fig. 32). A molecular model was built based on the X-ray structure of (1c)$_3$⊃[3a]$^{2+}$ and 2D NMR results of (1a)$_6$⊃([2a]Cl), implying that the distorted hexameric capsule formed an open structure around the guest (Supplementary Fig. 16). As the capsule structure of resorcin[4]arene hexamer (1a)$_6$ is stabilized by multiple hydrogen-bonds with the eight contained water molecules, the distorted hexameric capsule structure of (1a)$_6$⊃([2a]Cl) should also be supported by the incorporation of several water molecules[42].

### The effect of ligand bulkiness and counter anions

In this symmetry-breaking assembly, steric bulkiness of the Ir complex salt plays a crucial role in forming the remarkably low-symmetric structure. When the sterically less demanding Ir complex [2b]Cl was used as a guest instead, containing methyl (Me) substituents on the bpy ligand instead of tertiary butyl ($^t$Bu) groups as in [2a]Cl, the $^1$H NMR study revealed that encapsulation of [2b]Cl within the resorcin[4]arene hydrogen-bonded capsule proceeded to give a more common type of host–guest complex, showing a higher averaged symmetry due to the larger mobility of the smaller guest inside the cavity (Supplementary Fig. 22). On the other hand, an Ir complex having even larger substituents (phenyl groups) on the bpy units, [2c]Cl, did not yield any host–guest complex in solution, as judged from a $^1$H NMR study (Supplementary Fig. 25). This can be rationalized with the increased size of the Ir complex which is too large to be encapsulated within the hexameric capsule and the shape of [2c]$^+$ does not template any other resorcin[4]arene oligomer. Therefore, these results demonstrate that this symmetry-breaking assembly is a size- and shape-specific molecular recognition phenomenon in solution.

In addition, an anion-dependent behavior was observed distinctly. When the counter anion Cl$^-$ was replaced with NO$_3^-$, host–guest complexation also proceeded, as confirmed by NMR studies (Supplementary Fig. 17). Notably, the proton signals in 1a are severely broadened and the molecular symmetry of the guest stays $C_2$ on the NMR time scale after encapsulation. It implies that the NO$_3^-$ anion weakens the hydrogen-bond network in the capsule and thus increases the flexibility of 1a, producing a symmetric cavity. On the other hand, large anions (PF$_6^-$, BF$_4^-$, ClO$_4^-$, and OTf$^-$) did not show host–guest formation (Supplementary Fig. 21). The typical hexameric capsule can accommodate a cationic guest molecule as its ion-pair through hydrogen-bonds between weakly acidic H$_2$O molecules in the capsule framework and hydrogen-bond accepting anions[31, 33]. Thus, the anions also participate in the self-assembly process and are able to control the flexibility of the supramolecular structure[43]. Recently, it has also been reported that hydrogen-bonds between a resorcin[4]arene capsule and Cl anions afforded an anion-sealed capsule to produce a larger internal cavity in weakly anion-solvating solvents, such as THF and benzene, while not being stabilized by ammonium cations in chloroform[44]. For our system, we propose that the formation of an enlarged capsule, capturing a Cl anion in the hydrogen-bonding network, is likewise possible owing to the template effect of the rather large Ir complex guest. Apparently, this process is favored over ion-pair formation of the guest salt within the neutral hexameric capsule. Although the solution structure of this symmetry-breaking assembly still remains uncertain, these results definitely reveal that the cooperative effect among the guest cation, Cl anion, and hydrogen-bonded hosts plays a crucial role to form this remarkable assembly.

### Encapsulation-Induced Emission Enhancement (EIEE)

The photoluminescent properties of the Ir complex guests, originating from metal-to-ligand charge transfer ($^3$MLCT) and triplet ligand-to-ligand charge transfer ($^3$LLCT) excited states (Supplementary Fig. 50)[45], are significantly modulated after encapsulation within the hydrogen-bonded capsule as compared with their free forms in solution, owing to drastic changes of local environments around the guests by formation of highly aggregated structures[46]. The emission spectrum of encapsulated [2a]Cl ($\lambda_{em} = 564$ nm) shows a large blue shift ($\Delta\lambda_{em} = -19$ nm) from that of free [2a]Cl ($\lambda_{em} = 583$ nm) after mixing of 1a (60 µM, 6 eq.) and heating the mixture at 50 °C for 1 min (Fig. 4). In addition to the blue shift of the emission, higher emission quantum yields ($\Phi$) and longer emission lifetimes ($\tau_{ave}$) were observed, rising from 40% and 510 ns to 63% and 1020 ns, respectively. Further addition of 1a (300 µM, 30 eq.) into the solution of [2a]Cl enhances the photoluminescent properties ($\lambda_{em} = 562$ nm, $\Phi = 73\%$, $\tau_{ave} = 1090$ ns) of (1a)$_6$⊃([2a]Cl) even further, because the supramolecular structure is formed via two coupled equilibrium processes (self-assembly of 1a and encapsulation of [2a]Cl), and therefore a high concentration of 1a favors the formation of the host–guest complex. The non-radiative rate constants ($k_{nr}$) of the photoluminescence were significantly reduced by addition of 1a, indicating that the tight host–guest complex formation effectively suppressed the molecular dynamics of [2a]$^+$ and showed EIEE behavior. It is noted that this EIEE behavior was also observed with guests [2a]NO$_3$ and [2b]Cl (Supplementary Fig. 37 and 39), which were also entrapped within the symmetric hexameric capsule (1a)$_6$, while the photoluminescent properties of [2a]PF$_6$ and [2c]Cl were not significantly altered since they cannot fit inside the cavity of the resorcin[4]arene oligomer (Supplementary Fig. 38 and 40).

### Symmetry-breaking triggers modulation of chiroptical properties

Although these host–guest complexes showed an EIEE behavior, the symmetry-breaking effect in (1a)$_6$⊃([2a]Cl) was not observed clearly in the luminescent properties as discussed above. Therefore, we next

investigated symmetry-dependent photophysical properties of the Ir complexes. The chiral-at-metal Ir complexes are obtained as a 1:1 racemic mixture of Δ- and Λ-stereoisomers in a typical synthetic manner. The optical resolution of the mixture can give enantiopure complexes showing CD and CPL properties, which directly reflect the structural information in the molecular system[47].

The enantiopure Ir complexes Δ-[2]Cl and Λ-[2]Cl were obtained by auxiliary-mediated asymmetric synthesis according to established methods (Supplementary Fig. 1)[48, 49]. The CD spectra of Δ- and Λ-[2]Cl clearly showed mirror image spectra irrespective of their counter anions, indicating that the enantiopure Ir complexes possessed opposite metal-centered configuration (Supplementary Fig. 45). Upon

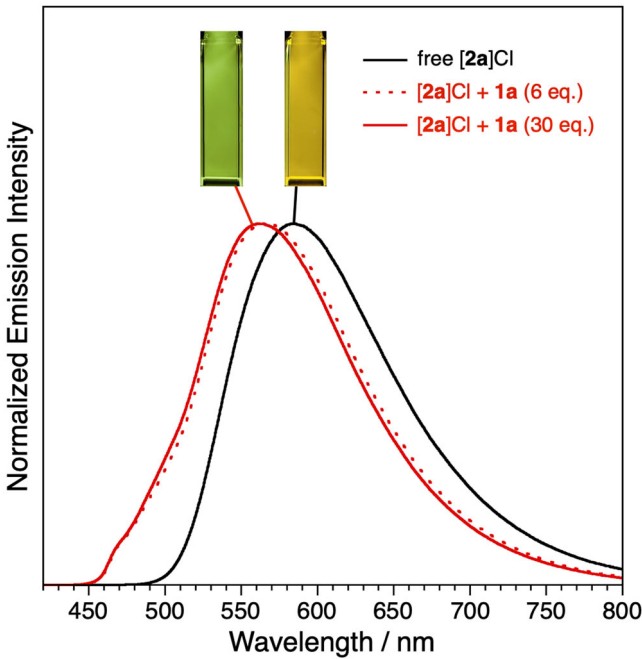

**Fig. 4 | Normalized emission spectra of [2a]Cl (10 μM) with various amounts of resorcin[4]arene monomer 1a (0, 60, 300 μM).** The solutions containing [2a]Cl and 1a were heated at 50 °C for 1 min before the measurements ($\lambda_{ex}$ = 400 nm).

addition of 1a into the solution containing Δ- or Λ-[2a]Cl, respectively, gradual spectral changes were observed in the CD spectra of the Cl salts (Fig. 5a), while those of the PF$_6$ salts did not change significantly (Supplementary Fig. 48a). As the PF$_6$ salt is not encapsulated within the capsule as confirmed by NMR studies, the spectral change in Fig. 5a unambiguously originates from the formation of the host–guest structure. In particular, the cotton effect around 420 nm, which derives from the chiral Ir complex, gradually blue shifts upon addition of 1a. These results therefore indicate that encapsulation within the hydrogen-bonded capsule can slightly alter the chiroptical property of the chiral guests. Since resorcin[4]arene absorbs light only in high energy UV region (<320 nm), chiral information transfer on the capsule might be observed due to the encapsulation of a chiral guest[50]. However, it should be noted that these results are still indirect evidence of the symmetry-breaking effect upon the host–guest assembly, because the CD spectra of smaller chiral Ir complex Δ- and Λ-[2b]Cl were also altered upon addition of 1a into the solution (Supplementary Fig. 49a).

Stronger evidence of the symmetry-breaking clearly derives from the circularly polarized luminescence (CPL) studies. CPL analysis allows to probe specifically the chiral Ir complexes[47], since the resorcin[4]arene hydrogen-bonded capsule are not emissive. Comparison between free and encapsulated complexes in hexameric hosts provides a unique perspective on how the symmetry-breaking affects the chiroptical properties of the system. The Δ- and Λ-configured complexes [2a]Cl in degassed CHCl$_3$ upon excitation at 350 nm clearly show right- and left-handed CPL effects ($\lambda_{max}$ = 580 nm), respectively, with a dissymmetry factor $g_{lum}$ = 2 × 10$^{-4}$ (Fig. 5b). After addition of resorcin[4]arene 1a into the solution and heating at 50 °C for 1 min, EIEE effects in the CPL properties are observed as a consequence of the host–guest formation as well as changes in the chirality-independent luminescent properties, including blue shift of the emission peak ($\lambda_{max}$ = 530 nm) and enhancement of the emission intensity in the CPL spectra (Supplementary Fig. 46). Surprisingly, the dissymmetry factor $g_{lum}$ of the CPL increases fourfold (8 × 10$^{-4}$) by the symmetry-breaking host–guest formation event (Fig. 5b). According to the physical theory, the $g_{lum}$ value is related to the ED and MD transitions moments in the radiative process (emission) as follows:

$$g_{lum} = 4|\boldsymbol{\mu}||\boldsymbol{m}|\cos\theta/(|\boldsymbol{\mu}|^2 + |\boldsymbol{m}|^2) \qquad (1)$$

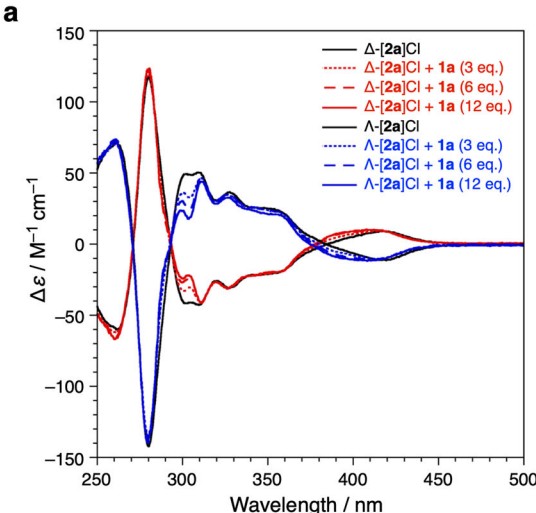

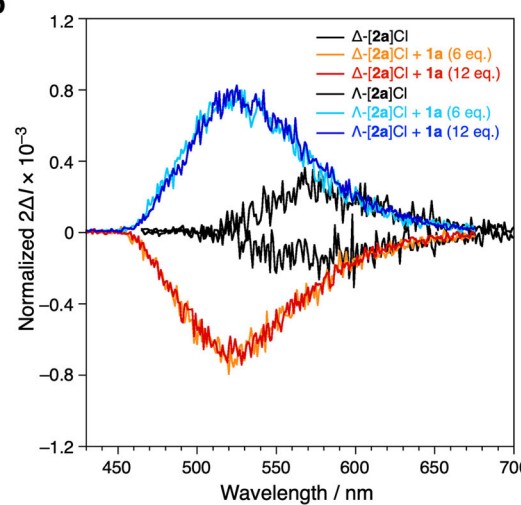

**Fig. 5 | Chiroptical properties of the complexes. a** Circular dichroism (CD) spectra (10 μM, CHCl$_3$, RT) of Δ- and Λ-[2a]Cl in the presence of resorcin[4]arene 1a (0, 30, 60, 120 μM). **b** Normalized circularly polarized luminescence (CPL) spectra (50 μM, CHCl$_3$, RT) of enantiomerically pure Ir complex [2a]Cl (50 μM) with resorcin[4]arene monomer 1a (0, 300, 600 μM), which was obtained from division of the 2Δ*I* curve by the maximum of the emission intensity. All samples solutions containing [2a]Cl and 1a were heated at 50 °C for 1 min before the measurements.

where $\mu$ is the transition ED moment, $m$ is the transition MD moment, and $\theta$ is the angle between $\mu$ and $m$[51]. As the free Ir complexes possess $C_2$ molecular symmetry, two phenylpyridinato ligands at the Ir metal center and two pyridine moieties in the bipyridine ligand are equivalent. On the other hand, after the symmetry-breaking assembly, the molecular symmetry in the supramolecular system lowers to point group $C_1$, leading to desymmetrization of the chemical environment of the organic ligands (Fig. 2a). Therefore, after vibrational relaxation of the excited state structure upon sensing the asymmetric environment originated from symmetry-breaking assembly, the vectors of the transition ED moment ($\mu$) and the transition MD moment ($m$) in the host–guest assembly should differ from those of the free Ir complex (Supplementary Fig. 51). In fact, the dissymmetry factor $g_{lum}$ of the CPL properties of chiral [2a]NO$_3$ and [2b]Cl did not change significantly upon addition of 1a, although the EIEE effect in the CPL spectra of chiral Ir complex salts, that is the blue shift of the emission peak and increase of the emission intensity in the CPL spectra, was observed similarly to the case of [2a]Cl with 1a (Supplementary Fig. 47 and 49). This is probably because these two Ir complex salts gave host–guest complexes without forming the symmetry-breaking assembly, supported by matching observations in the NMR studies. Thus, these CPL results are a clear evidence of the symmetry-breaking in the supramolecular assembly with guest [2a]Cl[52]. A careful design of the supramolecular system allows to couple the EIEE effect with an enhancement of the chiroptical properties, providing a new strategy for tuning the efficiency of CPL emitting materials.

This study showed the formation of $C_1$-symmetric supramolecular coordination complexes insulated by resorcin[4]arene oligomers in solution. Although resorcin[4]arenes typically form symmetric hydrogen-bonded capsules, encapsulation of shape-specific coordination complexes caused a structural change of the symmetric host frameworks to afford a distorted hexameric capsule in solution, creating asymmetric environments inside the oligomer, where the organic ligands of the guests are located in nonequivalent positions. The structures of the symmetry-breaking assemblies in this supramolecular system were elucidated by extensive 2D NMR spectroscopic analyses. CD and CPL measurements in solution revealed that the encapsulated chiral metal complexes are strongly affected by the surrounding asymmetric environment. The synergistic effect of weak interactions between the components is a key to construct these $C_1$-symmetric assemblies, because the combination of multiple weak interactions can provide enough enthalpic gain to overcome the large entropy loss of $C_1$-symmetric products in the multi-component self-assembly. As the distribution of the weak binding forces is basically controllable by a tuning of reaction conditions, thus, the present strategy might provide a useful methodology to bring more complexity and flexibility into self-assembled nano systems. The preparation of relatively large anisotropic, yet defined supramolecular architectures based on chiral components bears potential to develop new applications in areas such as molecular chiroptical materials, selective luminescent diagnostics and homogeneous enantioselective photocatalysis.

## Methods

The $^1$H and DOSY NMR spectra were obtained on a 500 MHz Varian NMR System 500PS spectrometer. 2D NMR measurements were conducted by using a Bruker Avance III 800 spectrometer. Chemical shifts are reported as values in p.p.m. relative to tetramethylsilane ($\delta = 0$) in deuterated solvents. Electrospray ionization (ESI), tandem mass, and ion mobility spectrometry were performed on a Bruker ESI timsTOF mass spectrometer. UV/Vis absorption spectra, emission spectra, emission lifetime decay profiles, and emission quantum yields were obtained on a Jasco V-560 spectrophotometer, Jasco FP-6500 spectrofluorometer, a Hamamatsu Photonic Absolute PL Quantum Yield Measurement System C9920-02 integrating sphere

and a PMA-12 multichannel photodetector (excitation wavelength = 420 nm), and a Hamamatsu C11200 streak camera as a photodetector by exciting at 355 nm using a nanosecond Q-switched Nd:YAG laser (Continuum Minilite), respectively. The CD spectra were collected on a Jasco J-75W CD spectrometer. The CPL spectra were recorded on a JASCO CPL-300 spectrophotometer, equipped with a 150 W Xenon lamp and a PMT detector. All CPL measurements were carried out at room temperature, whose spectra correspond to an average of 10 scans with a fixed excitation bandwidth of 25 nm, a fixed emission bandwidth of 15 nm and an integration time of 1 s. All photophysical measurements were performed under deaerated conditions after N$_2$ bubbling for 30 min.

## Data availability

The authors declare that the data supporting the findings of this study are available within this article and supplementary information files, as well as from the corresponding authors upon request. The crystallographic data for the structure reported in this study have been deposited in the Cambridge Crystallographic Data Centre, under accession number 2152087, and it can be obtained free of charge from the Cambridge Crystallographic Data Centre via http://www.ccdc.cam.ac.uk/data_request/cif.

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

## Acknowledgements

This research was supported by JSPS KAKENHI Grant Numbers JP19K15589, JP19H04587, JP20H05231, JP22H04554 (S.H.), JP19H04569 (T.Y.), and JP17K14463, JP20K05542 (K.U.), and by the JGC-S Scholarship Foundation, the Noguchi Institute, the Ogasawara Foundation for the Promotion of Science & Engineering, the Izumi Science and Technology Foundation, the Takahashi Industrial and Economic Research Foundation, and the Nagasaki University WISE programme

(S.H.). S.H. and E.S. thank the JSPS Program for Advancing Strategic International Networks to Accelerate the Circulation on Talented Researchers. This work was a result of using research equipment shared in MEXT project for promoting public utilization of advanced research infrastructure (Program for supporting introduction of the new sharing system) Grant Number JPMXS0422500320 and the Advanced Research Infrastructure for Materials Nanotechnology (ARIM) Japan (Project code JPMXP1222JI0014). This work was funded by the Deutsche Forschungsgemeinschaft (DFG, German Research Foundation) under Germany's Excellence Strategy – EXC 2033 – 390677874 – RESOLV. We are grateful to Junko Nagaoka and Laura Schneider for assistance in the X-ray and ESI-MS analyses, respectively.

## Author contributions

This study was designed by S.H. Synthesis and measurements of the non-chiral compounds were performed by S.H., T.H., and E.S. 2D NMR analysis was conducted by T.Y. CPL studies were performed by J.T. The chiral compounds were prepared by S.H and E.M. The manuscript was co-written by S.H., T.Y., J.T., Y.A., G.C., and K.U. All authors discussed the results and commented on the manuscript.

## Competing interests

The authors declare no competing interests.
