## [Peer Review File · Nature Communications]

Symmetry-Breaking Host–guest Assembly in a Hydrogen-bonded Supramolecular SystemREVIEWER COMMENTS

Reviewer #1 (Remarks to the Author):

The paper from Horiuchi, Clever, and Umakoshi reports the formation of new encapsulation complexes between a resorcinarene capsule and luminescent coordination compounds as guests. The unique feature of these complexes is that they have reduced symmetry and exhibit enhanced emission (including circularly polarized emission).

Although resorcin[4]arene hexameric capsules have been known for more than 20 years and their encapsulation properties have been extensively studied over those years, they still inspire chemists in various fields. This is mainly due to their unusual effectiveness in encapsulation of positively charged guest molecules that result in remarkable applications (e.g. in catalysis). In the paper under review, the authors present the encapsulation of cationic coordination complexes inside a distorted resorcinarene capsule. The paper constitutes a development of an initial discovery made by the authors made several years ago. The novelty in the present study involves the application of guest molecules with bulky side substituents that, themselves, have reduced symmetry and, due to their bulkiness, induce further distortion of the symmetry of the encapsulation complexes. Enantiomerically pure versions of guest molecules have also been used to form chiral complexes, and the influence of encapsulation on their ECD and CPL spectra has been discussed. Although several aspects of the encapsulation process, such as the enhancement of emission cationic coordination guests inside hexameric resorcin[4]arene capsules and anion dependence, have already been reported, the current symmetry-broken system brings a substantial development to the field. Breaking the symmetry of the capsules and obtaining chiral capsules are desired features for prospective applications in enantioselective catalysis, and controllable ways to achieve it remain mostly unknown. The current system is an important step towards this goal, however, with the obvious limitation coming from the fact that the cavity is occupied.

I think the subject of the paper is interesting, but I have numerous experimental doubts concerning the main findings reported in the paper. It is also difficult to envision further development of the system. Therefore, I recommend a major revision that involves many additional control experiments and substantial rewriting to put the results in a wider perspective.

1. My main objective on merits concerns the identity and homogeneity of the complexes that are claimed to be C₁-symmetric. The authors deduced the reduced symmetry and pentameric stoichiometry of the encapsulation complexes, based on the number of signals in the ¹H NMR spectrum and the ESI MS spectrum. However, the ESI MS spectrum clearly shows the presence of a mixture of aggregates, while the NMR spectrum was not fully assigned (so many signals remain). The DOSY technique has a limited resolution and may not reflect the differences between, for example, the pentameric capsule and the hexameric one. Therefore, I have doubts if the observed ¹H NMR spectrum really reflects the claimed species or is simply a mixture of complexes of different stoichiometry. Since this is the main finding of this paper, I think it should be unambiguously proved. Therefore, I recommend that the authors perform a full assignment of the ¹H NMR spectrum to prove that the signals originate from a single species. I know that the spectrum is complicated, but it is still sharp and well-resolved so with the help of HMBC, and NOESY/ROESY (maybe also ¹⁵N) it should be possible. The authors should also check the time dependence of spectra (e.g. after longer heating or after several days, for all complexes) to see if the equilibrium has been reached.

2. There are recent reports that discuss the interactions of resorcinarenes with anions (including those of the authors, but not only), which play an important role in self-assembly and capsule dissociation. Here, it is also clear that self-assembly is anion-dependent. The authors should discuss this in the context of those recent findings and perform additional experiments that will explain the role of anions (which may induce dissociation of the capsules).

3. The X-ray structure serves as a hint for a deduction of the possible structure in the solution, which is a correct approach. Here the formation of capsular encapsulation complexes based on this structure is postulated, which, however, requires inter-resorcinarene interactions. These interactions are not discussed at all, and they are also not visible in the provided picture.

4. I don't understand the large entropy difference between the formation of symmetric vs. asymmetric self-assembled species composed of the same number of components (and the same rigidity). Could you please provide a citation/rationale for this claim?

5. Discussion of diffusion coefficients - The authors should discuss D/D rather than $\Delta(D)$. In light of the Stokes-Einstein equation, $\Delta(D)$ has no physical meaning.
6. The authors claim, based on the spectrum Suppl.11, that the 2b complex is a “more common” hexameric capsule. The spectrum is very complex, and the signals are broad. I think this statement is not justified.
7. The $g(\text{lum})$ values should be put in a literature context – are the differences meaningful, and what is the error?
8. Differences in CD spectra are quite small, they may have been caused by the removal of the anion from the coordination sphere around the metal. The same can be true for the emission shift/enhancement and for changes in emission lifetimes. The authors should provide control experiments to explain the role of anions.
9. The ^1NMR spectra of enantiomerically pure guests with resorcin[4]arenes should be checked for identity with the spectra with racemic guests. In principle, they should be identical, but this has to be double-checked because the system is complex and the possibility of forming higher aggregates exists.
10. Clarity of presentation. The chemical structures discussed in this work should be clearly illustrated in the main text (it is hard to find in the text the differences between 1a/1b or 2a/2b/2c). The abstract is too general (it does not state the type of guests used). There are parts of the paper that should be rewritten because they are difficult to follow (e.g., paragraphs that start from “Coordination-driven...” or “Encapsulation studies...”).

Reviewer #2 (Remarks to the Author):

This manuscript by Horiuchi et al reports on the host-guest studies for the hydrogen-bonded resorcin[4]arene (1a-c) capsules and several Ir complexes (2a-c and 3a-c). The authors claimed that symmetry-breaking happened from the well-known Oh-symmetric hexameric capsule to “defined” C1-symmetric host-guest supramolecular structures. Then the encapsulation-induced chiroptical changes of the Ir complexes were studied, where increase in both emission quantum yield, lifetime, and dissymmetry factor ($g(\text{lum})$) were measured upon inclusion by the supramolecular capsules. The results are interesting, especially concerning the late photo-physical sections. However, this reviewer has not been fully convinced on the structural characterizations part concerning the true host-guest species in solution, due to the lack of enough experimental evidence. The potential publication of this manuscript can only be considered after the following serious problems addressed carefully by the authors:

- (1) The formation of the $([2a]@1a)_5$ host-guest complex is based solely on ESI-MS measurement. This is not acceptable concerning that the solid structure of $[3a]@1c)_3$ has been clearly evidenced by X-ray.
- (2) While the authors tentatively assigned the guest signals for the $^1\text{H NMR}$ of the $([2a]@1a)_5$ complex. Throughout the main-text and SI, NO assignment on the NMR were performed for the host-guest complexes. I understand it is a challenging task concerning the low-symmetry of the complex. However, without such assignment, the claim for the 1:5 ratio for the $2a@1a)_x$ complex are not acceptable.
- (3) If the authors uphold on the claimed “exact” host-guest species in solution, then binding constants for should be discussed.

Last but not the least, if the structural characterizations for the host-guest complexes turned out to be mission-impossible. I would recommend the authors use a vague description, i.e. oligomeric $[1a]_n$ for the their capsule structures, in a carefully revised manuscript. Referring to the papers reported by the Yoshizawa group, without such coercive structural claims on the host-guest complexes, I think the results obtained here will still stand out for an important contribution to the field.

Reviewer #3 (Remarks to the Author):

This is a very interesting manuscript that describes dynamic behavior of the resorcinarene capsule. The idea is that the capsule undergoes symmetry breaking from Oh to less symmetrical C1 symmetry. There is a wealth of characterization techniques that are supportive of the observations. Overall, I believe that the work fits the scope of Nature Communications.

With that being said, the claims are a bit vague. The idea of 'symmetry breaking' in my mind is that an entity of a given formula undergoes a change a reduction in symmetry and remains the same formula. The authors do make such an observation with the smaller Ir complex. However, a majority of the discussion related to the capsule undergoing a change in structure to a pentamer and trimer. This is dynamic behavior that is important, as the authors note, but it is not symmetry breaking. I believe that the authors need to be clearer on what is undergoing the symmetry breaking and clearly indicate that the other phenomena are not symmetry breaking (although the symmetries of the new entities are certainly lower). It should also be emphasized that the capsule itself is also chiral (I432), which I believe is an important consideration.

Response to referees

REVIEWER REPORT(S):

Reviewer 1: Comments to the Author

Overall: The paper from Horiuchi, Clever, and Umakoshi reports the formation of new encapsulation complexes between a resorcinarene capsule and luminescent coordination compounds as guests. The unique feature of these complexes is that they have reduced symmetry and exhibit enhanced emission (including circularly polarized emission).

Although resorcin[4]arene hexameric capsules have been known for more than 20 years and their encapsulation properties have been extensively studied over those years, they still inspire chemists in various fields. This is mainly due to their unusual effectiveness in encapsulation of positively charged guest molecules that result in remarkable applications (e.g. in catalysis). In the paper under review, the authors present the encapsulation of cationic coordination complexes inside a distorted resorcinarene capsule. The paper constitutes a development of an initial discovery made by the authors made several years ago. The novelty in the present study involves the application of guest molecules with bulky side substituents that, themselves, have reduced symmetry and, due to their bulkiness, induce further distortion of the symmetry of the encapsulation complexes. Enantiomerically pure versions of guest molecules have also been used to form chiral complexes, and the influence of encapsulation on their ECD and CPL spectra has been discussed. Although several aspects of the encapsulation process, such as the enhancement of emission cationic coordination guests inside hexameric resorcin[4]arene capsules and anion dependence, have already been reported, the current symmetry-broken system brings a substantial development to the field. Breaking the symmetry of the capsules and obtaining chiral capsules are desired features for prospective applications in enantioselective catalysis, and controllable ways to achieve it remain mostly unknown. The current system is an important step towards this goal, however, with the obvious limitation coming from the fact that the cavity is occupied.

I think the subject of the paper is interesting, but I have numerous experimental doubts concerning the main findings reported in the paper. It is also difficult to envision further development of the system. Therefore, I recommend a major revision that involves many additional control experiments and substantial rewriting to put the results in a wider perspective.

Answer: We appreciate the reviewer's deep insight and valuable comments for this supramolecular system. We would carefully address the reviewer's comments and suggestions as follows.

Comment 1: My main objective on merits concerns the identity and homogeneity of the complexes that are claimed to be C₁-symmetric. The authors deduced the reduced symmetry and pentameric stoichiometry of the encapsulation complexes, based on the number of signals in the ¹H NMR spectrum and the ESI MS spectrum. However, the ESI MS spectrum clearly shows the presence of a mixture of aggregates, while the NMR spectrum was not fully assigned (so many signals remain). The DOSY technique has a limited resolution and may not reflect the differences between, for example, the pentameric capsule and the hexameric one. Therefore, I have doubts if the observed ¹H NMR spectrum really reflects the claimed species or is simply a mixture of complexes of different stoichiometry. Since this is the main finding of this paper, I think it should be unambiguously proved. Therefore, I recommend that the authors perform a full assignment of the ¹H NMR spectrum to prove that the signals originate from a single species. I know that the spectrum is complicated, but it is still sharp and well-resolved so with the help of HMBC, and NOESY/ROESY (maybe also ¹⁵N) it should be possible. The authors should also check the time dependence of spectra (e.g. after longer heating or after several days, for all complexes) to see if the equilibrium has been reached.

Answer: We really thank the reviewer for the kind considerations. We measured again several 1D and 2D high-resolution NMR spectra for a full assignment of the C_1 symmetric host-guest complex. As a result, we could assign all peaks in the ^1H NMR spectrum and revealed the 6:1 stoichiometry of the host-guest complex in solution by monitoring the integral ratio of the peaks, suggesting that the resorcin[4]arene capsule possesses a distorted “hexameric” form rather than the “pentameric” structure after the symmetry-breaking assembly. We include the new NMR results in the revised manuscript and explain them here as well.

In the NMR analysis, the ^1H - ^{13}C HSQC measurement is the most powerful technique for the full assignment of ^1H NMR spectrum. Thus, we carefully prepared new samples and measured them using high-resolution 800 MHz NMR spectroscopy through the Advanced Research Infrastructure for Materials Nanotechnology (ARIM) Japan (Project code JPMXP1222JI0014). The new ^1H - ^{13}C HSQC spectrum shown in Figure A1 clearly shows 24 cross peaks originated from aromatic protons of resorcin[4]arene molecules from 5.0 to 7.0 ppm. In addition to the HSQC results, the integral ratio of the peaks in the ^1H NMR spectrum suggests that the resorcin[4]arene capsule still is formed as a hexameric structure around the Ir complex guest. Thus, we can confirm that the six resorcin[4]arene units are located in inequivalent conditions, forming a distorted, yet defined, hexameric capsule with a defined C_1 symmetric structure on the NMR time-scale.

Figure A1. Enlarged ^1H - ^{13}C NMR spectrum (CDCl_3 , 25 $^\circ\text{C}$) around aromatic protons **1a** in host-guest complex $[(\mathbf{1a})_6][\mathbf{2a}]\text{Cl}$. The 24 cross peaks derived from the resorcin[4]arene marked with the magenta filled circle (●) were observed. The signals of resorcin[4]arenes showing strong NOE with the ^tBu proton of encapsulated $[\mathbf{2a}]^+$ were marked in a square. This result suggests that six resorcin[4]arene units form a defined C_1 symmetric hexameric capsule.

We obtained further structural aspects of the C_1 symmetric assembly by VT-HSQC measurement at 50 $^\circ\text{C}$. The HSQC spectrum recorded at 50 $^\circ\text{C}$ clearly showed significant broadening and disappearance of two broad cross-peaks assigned to resorcin[4]arene units, while other peaks were still sharp and slightly shifted (Figure A2). These results suggest that two in six resorcin[4]arene units are relatively flexible around the Ir guest, compared with

other four resorcin[4]arene units. In ^1H - ^1H NOESY measurement, 8 cross peaks between ^tBu groups in the Ir guest and two in the four rigid resorcin[4]arene units were clearly detected (Supplementary Fig. 6). In addition, the guest proton 3 and 8 showed strong Rotating frame nuclear Overhauser effect (ROE) with the other two rigid resorcin[4]arene units, implying that the pyridyl moieties in ppy ligands of $[\mathbf{2a}]^+$ were covered by resorcin[4]arene units (Figure A3). This binding motif was partially visualized by the X-ray structure of the host-guest complex using the Ru complex analogue. Therefore we conclude that six resorcin[4]arene units are placed in an inequivalent environment around one Ir complex cation. Time dependent NMR studies were also performed for all host-guest complexes. The NMR results showed no change of the signals, suggesting that the encapsulation reaches thermodynamic equilibrium within 1 h upon heating the solution at 50 °C. We revised the main text to add a new section about the solution structure of the distorted hexameric capsule determined by these NMR studies.

Figure A2. ^1H - ^{13}C HSQC spectrum of $(\mathbf{1a})_6\supset([\mathbf{2a}]\text{Cl})$ recorded at 50 °C (black), superimposed with the spectrum at 25 °C (red). The cross peaks marked with black and blue encircled numbers were assigned to the protons of free $[\mathbf{2a}]^+$ and encapsulated $[\mathbf{2a}]^+$, respectively.

Figure A3. ROESY spectrum (800 MHz, CDCl_3 , 25 °C) of $(\mathbf{1a})_6 \supset ([\mathbf{2a}]\text{Cl})$. The black and blue encircled numbers were assigned to the protons of free and encapsulated $[\mathbf{2a}]^+$, respectively. The signals of resorcin[4]arenes showing strong NOE with the ^tBu proton of $[\mathbf{2a}]^+$ were marked in a square.

We must explain the reason why the stoichiometry of the host–guest complex in the gas-phase, as observed by the ESI-MS study, showing the “pentameric” capsule as most intense signal, is different from the solution structure. We speculate that this is due to the lack of bridging H_2O molecules in the gas-phase analysis. As H_2O molecule plays a critical role to form the resorcin[4]arene hexameric capsule by multiple hydrogen-bonds connecting six resorcin[4]arene units in solution, the distorted hexameric capsule around the Ir complex cation is also expected to be stabilized by multiple hydrogen-bonds with water molecules (*J. Am. Chem. Soc.*, **2021**, *143*, 16419). However, in the gas-phase analysis, the water molecules are expelled during the measurement condition to give intense signals corresponding to the formula $[(\mathbf{1b})_n(\mathbf{2a})]^+$ ($n = 3\text{--}6$), lacking any bound water molecules. Therefore, the unusual hydrogen-bonded supramolecular structure could not be maintained and partial dissociation of the host was promoted in the ESI-MS study. The NMR study revealed that the distorted hexameric capsule contained two flexible resorcin[4]arene units in the capsule framework, suggesting one of the flexible units might be ejected very easily during the ESI-MS measurement. In order to avoid confusion about the stoichiometry of the assembly in solution, we moved the results of the gas-phase structure confirmed by ESI-MS measurements to the supplementary information. Similarly, the X-ray structure of the partial host–guest complexes was transferred to the SI, because the stoichiometry of the X-ray structure is also far different from that of the solution structure.

Again, we sincerely thank the reviewers for giving us the opportunity to revise the characterization of this remarkable assembly.

Comment 2: There are recent reports that discuss the interactions of resorcinarenes with anions (including those of the authors, but not only), which play an important role in self-assembly and capsule dissociation. Here, it is also clear that self-assembly is anion-dependent. The authors should discuss this in the context of those recent findings and perform additional experiments that will explain the role of anions (which may induce dissociation of the capsules).

Answer: According to reviewer's suggestion, we prepared the Ir complex salts having a variety of anions (NO_3^- , BF_4^- , ClO_4^- , and OTf^-) and checked formation of host-guest complexes and their photophysical properties. As the reviewer expected, a clear anion-dependent effect was observed in the NO_3^- salt, while the Ir complex salts with large anions, such as BF_4^- , ClO_4^- , and OTf^- , did not show a significant change in ^1H NMR spectrum as well as the results of PF_6^- . The NMR study using the NO_3^- salt also showed the formation of a host-guest complex in solution (Figure A4). Interestingly, in contrast to the result of Cl salt, the encapsulation of NO_3^- salt within the resorcin[4]arene capsule proceeded without heating the solution. Furthermore, the averaged proton signals of the capsule were observed and the molecular symmetry of the guest kept the C_2 symmetry in NMR time-scale even after the encapsulation. Therefore, these results suggest that the NO_3^- anion weakened the hydrogen-bond network of the capsule to increase flexibility of resorcin[4]arene units, producing a symmetric cavity, averaged on the NMR time-scale. Although the nitrate salt of the Ir complex showed encapsulation-induced emission enhancement (EIEE) behavior, the dissymmetry factor g_{lum} in the circularly polarized luminescence (CPL) study of the nitrate salt of chiral Ir complex did not change at all. This is in good agreement with the molecular symmetry of guest complex within the capsule, similarly to the results of smaller Ir complex salt $[\mathbf{2b}]\text{Cl}$. We add these discussions in the main text and experimental results in the supplementary information. We thank the reviewer for the fruitful suggestions.

Figure A4. ^1H - ^{13}C HSQC spectrum of $(\mathbf{1a})_6 \supset ([\mathbf{2a}]\text{NO}_3)$ recorded at 50 °C (black), superimposed with the spectrum of $(\mathbf{1a})_6 \supset ([\mathbf{2a}]\text{Cl})$ at 25 °C (red). The cross peaks marked with black and blue encircled numbers were assigned to the protons of free $[\mathbf{2a}]^+$ and encapsulated $[\mathbf{2a}]^+$, respectively.

Comment 3: The X-ray structure serves as a hint for a deduction of the possible structure in the solution, which is a correct approach. Here the formation of capsular encapsulation complexes based on this structure is postulated, which, however, requires inter-resorcinarene interactions. These interactions are not discussed at all, and they are also not visible in the provided picture.

Answer: As the reviewer pointed out, inter-resorcin[4]arene interactions in the X-ray structure is important to realize the binding motif in the supramolecular structure in the solid state. As we answered in comment 1, we would move the X-ray results in the SI. Thus, the new figures of the inter-resorcin[4]arene interaction were added in the SI (Supplementary Fig 30). This new figure clearly displayed that the resorcin[4]arene units were connected by the host–guest interaction and multiple hydrogen-bonds among the OH groups of **1c**, solvent molecules, and Cl anions. We thank the reviewer for the comments.

Comment 4: I don't understand the large entropy difference between the formation of symmetric vs. asymmetric self-assembled species composed of the same number of components (and the same rigidity). Could you please provide a citation/rationale for this claim?

Answer: We added citations as ref #21 and #22 in the main text. These references discuss the entropy-symmetry correlation. Molecular self-assembly is basically large entropy-loss process, because the number of molecules in the system drastically decrease via the self-assembly. However, highly symmetrical structures are entropically favored more than low symmetric assemblies, because the entropy loss to form a symmetric structure is suppressed when the molecular components are located at equivalent situations in the assembly and can not be distinguished each other, leading to information loss (entropy increase) of the components in the system. On the other hands, if low symmetric assembly is formed via self-assembly, all components can be discriminated, following with informational gain (entropy decrease). Thus, this is a main reason why highly symmetric (high entropy) structures are favored and obtained as a thermodynamic stable product in a molecular assembly.

Comment 5: Discussion of diffusion coefficients - The authors should discuss D/D rather than $\Delta(D)$. In light of the Stokes-Einstein equation, $\Delta(D)$ has no physical meaning.

Answer: According to the reviewer's suggestion, we changed the description of $\Delta(D)$ to $D_{(\text{host-guest})} / D_{(\text{free Ir})}$. According to the Stokes-Einstein equation, the D value is shown as

$$D = k_B T / 6\pi\eta r_H$$

where k_B is the Boltzmann constant, T the absolute temperature, η the viscosity and r_H the hydrodynamic radius of the species. Thus, the $D_{(\text{host-guest})} / D_{(\text{free Ir})}$ ratio is directly informative for molecular sizes ($r_{(\text{free Ir})} / r_{(\text{host-guest})}$) of two different species. In this case, the $D_{(\text{host-guest})} / D_{(\text{free Ir})}$ is approximately 0.34, indicating that the molecular size of host-guest complex is larger than that of free **[2a]⁺**. We thank the reviewer for the valuable comments.

Comment 6: The authors claim, based on the spectrum Suppl.11, that the **2b** complex is a “more common” hexameric capsule. The spectrum is very complex, and the signals are broad. I think this statement is not justified.

Answer: Thank you for the comments regarding ¹H NMR spectrum of the host-guest complex consisting of resorcin[4]arene and **[2b]⁺** having smaller substituents than **[2a]⁺**. Although we revised the stoichiometry of the host–guest complex using Ir complex cation **[2a]Cl**, we would still keep this description about the hexameric capsule of **[2b]Cl**. As the reviewer pointed out, the ¹H NMR spectrum of the host–guest complex (**1a**)₆⇌(**2b**)Cl showed complicated peaks. We assumed that this peak splitting and broadening were derived from symmetry

lowering of the “common” symmetric hexameric capsule rather than the formation of the “distorted” hexameric capsule. NMR spectra of host–guest complexes using Ir complex cations having different substituents at the bpy ligand were summarized in Figure A5. As we explained in the main text with several citations (ref 36–38), it has been reported that the symmetry lowering of self-assembled molecular hosts via a molecular recognition occurs, when the guest molecule is tightly trapped within the cavity of the host and the molecular tumbling of the guest became slow in NMR time-scale. In the case of host–guest complex with $[2b]^+$, the symmetry of the hexameric capsule would follow the guest symmetry, indicating that it should have C_2 or higher symmetry. The complicated and broadened NMR spectrum of $(1a)_6 \supset ([2b]Cl)$ is probably owing to slow rotation and site exchange of resorcin[4]arene units around $[2b]^+$. However, unlike $[2a]^+$, the host–guest interactions in $[2b]^+$ are not strong enough to freeze the molecular motions completely in NMR time-scale, giving the broad and complicated NMR spectrum (Fig A5c). Indeed, this NMR spectrum is similar to that of the smallest Ir complex cation $[Ir(ppy)_2(bpy)]Cl$ (Fig A5b). Importantly, the CPL study of $[2b]^+$ did not show significant change of dissymmetry factor g_{lum} in the presence and absence of the host. This CPL result strongly supports that the “typical” symmetric capsule is formed around $[2b]^+$ and a location of $[2b]^+$ in the symmetric cavity did not modulate electric dipole (ED) and magnetic dipole (MD) transition moments of the guest in the photo-excited state. Therefore, these results suggest that the symmetry lowering is observed only for the capsule without freezing molecular motions and the molecular symmetry of $[2b]^+$ remains in C_2 symmetry after encapsulation. As the molecular symmetry of host–guest complex $(1a)_6 \supset ([2b]Cl)$ is not C_1 symmetry, it should be emphasized again that this is not categorized in “symmetry-breaking” assembly.

Figure A5. a) Schematic representation of host–guest complexation using resorcin[4]arene **1a** and Ir complex cations having different substituents at the bpy ligand. b-d) ¹H NMR spectra (500 MHz, CDCl₃, r.t.) of host–guest complex using b) $[Ir(ppy)_2(bpy)]Cl$, c) $[Ir(ppy)_2(Me_2bpy)]Cl$, and d) $[Ir(ppy)_2(tBu_2bpy)]Cl$.

Comment 7: The g_{lum} values should be put in a literature context – are the differences meaningful, and what is the error?

Answer: We added a reference (ref 46) about the dissymmetry factor (g_{lum}) for the circularly polarized luminescence (CPL) study, as the reviewer suggested. According to the equation of the dissymmetry factor, the maximum g_{lum} value is 2, when $|\mu|$ and $|m|$ are equal and the directions of μ and m vectors are parallel or antiparallel ($\theta = \pm 1$). As summarized in the review written by Doistau et al. (*Front. Chem.*, **2020**, 8, 555), enantiopure Ir^{III} complexes show CPL properties with dissymmetry factor in the range of 10^{-4} to 5×10^{-3} . In our study, chloride salt of optically resolved di-*tert*-butylpyridine Ir^{III} complex, Δ - and Λ -[**2a**]Cl, showed g_{lum} value of 2×10^{-4} whose value fell into above range and the error was estimated to be 1×10^{-4} . Surprisingly, the g_{lum} value of the encapsulated chloride salt of the Ir^{III} complex (**1a**) \subset [(**2a**)Cl] became 8×10^{-4} , which was four times larger than that of free [**2a**]Cl, due to the symmetry-breaking assembly (Supplementary Fig 44b). In contrast to the results of [**2a**]Cl, Ir complex [**2a**]NO₃ and [**2b**]Cl did not show significant change of the g_{lum} values in the CPL properties after host-guest formation (Supplementary Fig 45b and 47b). These results strongly indicate that Ir complexes [**2a**]NO₃ and [**2b**]Cl were encapsulated within the resorcin[4]arene hexameric capsule without the symmetry-breaking phenomenon, as determined by NMR studies. Thus, we believe that the difference in the dissymmetry factor is notable and meaningful. We appreciate the reviewer's consideration.

Comment 8: Differences in CD spectra are quite small, they may have been caused by the removal of the anion from the coordination sphere around the metal. The same can be true for the emission shift/enhancement and for changes in emission lifetimes. The authors should provide control experiments to explain the role of anions.

Answer: As the reviewer pointed out, the CD spectra of the chiral Ir complexes showed only small changes in the presence of the capsule as a whole. The low sensitivity of CD spectra toward surrounding asymmetric environment (hexameric capsule with reduced symmetry) is more evident from the plots of g_{abs} values toward wavelength for [**2a**]Cl, [**2a**]NO₃ and [**2b**]Cl before and after encapsulation, showing that no enhancement of g_{abs} values occur upon encapsulation (Supplementary Figs. 44, 45 and 47). In our previous paper (ref 31), we have revealed that photo-absorption properties of the guest were insensitive to the host-guest formation, almost independent on the counter anions. The CD property is a kind of photo-absorption property of chiral compounds, which occurs within very short time (~ 1 fs) and reflects the ground state structures of Ir complex. On the other hand, CPL spectra reflect the information of the excited state structure of Ir complexes, implying that the Ir complex has enough time (~ 1 μ s) to sense the surrounding asymmetric environment during vibrational relaxation. Therefore the smaller changes in CD spectra seem to be reasonable. Recently, an interesting paper about the dynamic ion-pair reorganization in cyclometalated Ir complexes has been reported (*Nature Chem.* **2022**, 14, 746). Although our supramolecular system clearly showed the anion-dependent host-guest formation, the discussion on the ion-pair reorganization in the excited state is likely beyond the scope of the present work. Therefore, we would just conclude in this paper that the anions play an important role to form the host-guest structure and control the flexibility of the hydrogen-bonded capsule. Thanks to the reviewer's consideration, we have noticed that our supramolecular system might be suitable to provide new insights of dynamic ion-pair reorganization in cyclometalated Ir complexes. We sincerely appreciate the reviewer's consideration.

Comment 9: The ^1H NMR spectra of enantiomerically pure guests with resorcin[4]arenes should be checked for identity with the spectra with racemic guests. In principle, they should be identical, but this has to be double-checked because the system is complex and the possibility of forming higher aggregates exists.

Answer: As the reviewer suggested, we checked ^1H NMR studies of the host–guest complex using chiral Ir complexes. The results were shown in Figure A6 and added in the supporting information (Supplementary Fig 42). These results clearly showed that the symmetry-breaking assembly was independent on the chirality of the guest. We thank the reviewer for the valuable suggestion.

Figure A6. ^1H NMR spectra (500 MHz, CDCl_3 , r.t.) of host–guest complex $[(\mathbf{1a})_6] \supset [\mathbf{2a}]\text{Cl}$ obtained from a) racemic mixture of $[\mathbf{2a}]\text{Cl}$, b) Δ - $[\mathbf{2a}]\text{Cl}$, and c) Λ - $[\mathbf{2a}]\text{Cl}$.

Comment 10: Clarity of presentation. The chemical structures discussed in this work should be clearly illustrated in the main text (it is hard to find in the text the differences between 1a/1b or 2a/2b/2c). The abstract is too general (it does not state the type of guests used). There are parts of the paper that should be rewritten because they are difficult to follow (e.g., paragraphs that start from “Coordination-driven...” or “Encapsulation studies...”).

Answer: As the reviewer suggested, we added the chemical structures in Figure 1 and modified the abstract in order to catch the information about the compounds used in this study. The main text was also rewritten and divided by several paragraphs.

Reviewer 2: Comments to the Author

Overall: This manuscript by Horiuchi et al reports on the host-guest studies for the hydrogen-bonded resorcin[4]arene (1a-c) capsules and several Ir complexes (2a-c and 3a-c). The authors claimed that symmetry-breaking happened from the well-known Oh-symmetric hexameric capsule to “defined” C1-symmetric host-guest supramolecular structures. Then the encapsulation-induced chiroptical changes of the Ir complexes were studied, where increase in both emission quantum yield, lifetime, and dissymmetry factor (g_{lum}) were measured upon inclusion by the supramolecular capsules. The results are interesting, especially concerning the late photo-physical sections. However, this reviewer has not been fully convinced on the structural characterizations part concerning the true host-guest species in solution, due to the lack of enough experimental evidence. The potential publication of this manuscript can only be considered after the following serious problems addressed carefully by the authors:

Answer: We thank the reviewer for the consideration. We would address the reviewer’s questions by the additional explanations in order to improve the understandability of the results.

Comment 1: The formation of the $([2a]@(1a)_5)$ host-guest complex is based solely on ESI-MS measurement. This is not acceptable concerning that the solid structure of $[3a]@(1c)_3$ has been clearly evidenced by X-ray.

Answer: As we explained in the answer of comment 1 from the reviewer 1, the stoichiometry of the symmetry-breaking assembly is revised in a 6:1 host–guest complex, as revealed by additional NMR experiments, that is far different from the stoichiometry of its gas-phase and solid structures. The photophysical properties of the assembly were studied in the solution states. Thus, in order to avoid confusion about the stoichiometry of the assembly in solution, we would move the ESI-MS and X-ray results to the supplementary information.

We would explain again a reason why the stoichiometry of the assembly is depended on the measurement conditions. These supramolecular structures are formed by the concerted effect of weak binding forces, such as hydrogen-bond, electrostatic, cation- π and CH- π interactions, being reminiscent of natural system. In this paper, we clearly demonstrated that this bio-inspired supramolecular system affords a highly unusual aggregated structure having a remarkable complexity. Here, it should be mentioned that this supramolecular assembly is rather flexible and of dynamic nature, because individual associating forces are easily altered by changing reaction conditions. For example, in the NMR measurements, the host–guest complexes are embedded in a solvent environment during the measurement, making them structurally behave as individual host–guest complexes. In contrast, single crystals suitable for X-ray analysis are typically grown from a concentrated solution of the assemblies. Thus, in the crystalization process, not only the host-guest interactions but intermolecular interactions among the host-guest complexes effectively control the molecular alignments in the crystal growing process. As a result, the flexible supramolecular complexes show a remarkable structural variation from 6mer to 3mer dependent on the reaction condition. Such flexible behavior is also observed even in rigid supramolecular coordination cages (*Inorg. Chem.*, **2012**, *51*, 9574).

Although the X-ray structure of $[(1c)_3\supset 3a]^{2+}$ is not the same as the solution state owing to the flexible nature of this molecular system, this X-ray structure clearly explained the steric effect from two ^tBu groups in the guest, destabilizing the typical resorcin[4]arene hexamer. We believe that the NMR results, in particular the ¹H–¹³C HSQC measurement, deliver convincing structural information to assign a $(1a)_6\supset([2a]Cl)$ stoichiometry in solution. We thank the reviewer for the comments.

Comment 2: While the authors tentatively assigned the guest signals for the HNMR of the ([2a]@(1a)₅) complex. Throughout the main-text and SI, NO assignment on the NMR were performed for the host-guest complexes. I understand it is a challenging task concerning the low-symmetry of the complex. However, without such assignment, the claim for the 1:5 ratio for the 2a@(1a)_x complex are not acceptable.

Answer: As shown in the answer of the comment 1, we performed a series of additional measurements and now achieved full assignment of the signals in the ¹H NMR spectrum (Figure A1). In particular, the ¹H-¹³C HSQC measurement clearly showed that six resorcin[4]arene located at inequivalent situation around the Ir complex cation. Thus, we would revise the stoichiometry of the host-guest complex. We sincerely appreciate the reviewers for giving this opportunity to revise the stoichiometry of this symmetry-breaking assembly.

Comment 3: If the authors uphold on the claimed “exact” host-guest species in solution, then binding constants for should be discussed.

Answer: We thank you for the suggestion. Of course, we realize that binding constants are informative for quantitative discussion in a molecular recognition process. However, we suspect that this supramolecular system is difficult to discuss with binding constants, because the host-guest formation is based on several equilibria, such as self-assembly of resorcin[4]arene host, encapsulation of the guest and re-organization of the capsule around the guest. In addition, the encapsulation of guest within the resorcin[4]arene hexameric capsule requires heating. These issues would provide some analytical uncertainty for obtaining meaningful values of binding constants from the experiments. Furthermore, given the extreme complexity of the NMR spectra, titration data would not allow for the unambiguous quantification of the binding constant.

Comment 4: Last but not the least, if the structural characterizations for the host-guest complexes turned out to be mission-impossible. I would recommend the authors use a vague description, i.e. oligomeric [1a]_n for the their capsule structures, in a carefully revised manuscript. Referring to the papers reported by the Yoshizawa group, without such coercive structural claims on the host-guest complexes, I think the results obtained here will still stand out for an important contribution to the field.

Answer: We appreciate the reviewer for the kind consideration. Thanks to the several additional NMR experiments, we fully characterized the NMR signals and determined the stoichiometry of the host-guest complexes in solution. We believe that these new results will convince the reviewer’s considerations.

Reviewer 3: Comments to the Author

Overall: This is a very interesting manuscript that describes dynamic behavior of the resorcinarene capsule. The idea is that the capsule undergoes symmetry breaking from O_h to less symmetrical C_1 symmetry. There is a wealth of characterization techniques that are supportive of the observations. Overall, I believe that the work fits the scope of Nature Communications.

Answer: We thank the reviewer for the kind comment. We also believe that these results in this paper have a significance enough to catch great interests to the broad readership of Nature communications.

Comment 1: With that being said, the claims are a bit vague. The idea of 'symmetry breaking' in my mind is that a entity of a given formula undergoes a change a reduction in symmetry and remains the same formula. The authors do make such an observation with the smaller Ir complex. However, a majority of the discussion related to the capsule undergoing a change in structure to a pentamer and trimer. This is dynamic behavior that is important, as the authors note, but it is not symmetry breaking. I believe that the authors needs to be clearer on what is undergoing the symmetry breaking and clearly indicate that the other phenomena are not symmetry breaking (although the symmetries of the new entities are certainly lower). It should also be emphasized that the capsule itself is also chiral ($I432$), which I believe is an important consideration.

Answer: Thank you very much for the consideration. As the reviewer stated, when a compound having a particular formula shows a reduction of symmetry without any change in the chemical formula, such behavior can be categorized as "symmetry-breaking" phenomenon. In this paper, we observed symmetry reduction of host-guest complexes (O_h (hexameric capsule) and C_2 (guest) symmetric units) to the lowest C_1 symmetry via self-assembly in a molecular level, as confirmed by NMR measurement. All protons in the guest complexes were inequivalently observed in the NMR spectra, indicating that all the molecular units are located at chemically inequivalent positions via self-assembly to form the "defined" C_1 symmetric assembly. We would emphasize in this paper that this symmetry reduction of the host-guest complexes to give C_1 symmetric assembly is "symmetry-breaking assembly". In particular, the new results of the additional experiments suggested by reviewers led to revise the chemical formula of the host-guest complex from $[(1a)_5\supset(2a)Cl]$ to $[(1a)_6\supset(2a)Cl]$. This revised stoichiometry corresponds to the symmetry-breaking phenomena that this reviewer considered.

In addition, as reported by MacGillivray and Atwood in 1997, resorcin[4]arene hexameric capsule crystallizes in chiral $I432$ space group. It suggests that the hexameric capsule creates a chiral cavity inside the capsule in the crystalline state through rapid recombination of multiple hydrogen-bonds in the capsule framework. On the other hand, in contrast to the chiral assembly in the crystalline state, the non-chiral resorcin[4]arene hexameric structure is formed due to the labile and dynamic character of the hydrogen-bonding assembly in solution, as confirmed by NMR and CD studies. However, the large Ir complex cation can induce effective template effect to fix the chiral hydrogen-bonding structure. In fact, the CD spectra of the inclusion complexes of chiral $[2a]Cl$ and $[2b]Cl$ might show chiral information transfer in high energy UV region (<320 nm). It is definite that the symmetry-breaking assembly observed for this system owe to the flexibility of the hydrogen-bonded capsule.

REVIEWER COMMENTS

Reviewer #1 (Remarks to the Author):

The authors have made a great effort to explain the structural aspects of the formation of the symmetric C₁ encapsulation complex of resorcin[4]arene with the luminescent Ir complex. In the revised version of the manuscript, many crucial issues have been clarified. The authors admitted their initial error in assigning the structure as pentameric and corrected their assignment to the hexameric capsule (based on the analysis of 2D NMR spectra). This seems to be a large structural difference; however, in fact, it has a minor effect on the general conclusion, which still holds. The authors addressed the questions and undoubtedly the quality of the article improved considerably. However, the paper still needs some substantial corrections and additional interpretations (major revision).

1. On the basis of the provided data for the complexation of Ir complexes containing various counterions, I think that the role of anions is still underestimated and not interpreted properly in this paper. Most likely, the anions participate in self-assembly, which is consistent with all the NMR data in this paper and is also related to recent findings (JACS 2022, 5350). For the resorcin[4]arene capsule that encapsulates the Ir-complex-Cl, the substantial downfield shift of OH groups is observed (compared to the free capsule and complexes containing other counterions). Therefore, anions may form hydrogen bonds with OH groups, rather than be co-encapsulated. This conclusion is in agreement with smaller changes observed during encapsulation of Ir-complex-NO₃⁻ (the NO₃⁻ anion, due to charge distribution, forms much weaker interactions with OH groups). The interpretation is also in agreement with the lack of encapsulation of Ir-complex-BF₄ or Ir-complex-PF₆, which contain anions that do not interact with the OH groups. The anions may be crucial for symmetry lowering, therefore I insist on a better discussion of the role of the anions in the paper. It is particularly relevant because changes in chiroptical properties observed for encapsulation also seem to depend on counterions.

2. When symmetry lowering is discussed, it is crucial to know if all signals belong to the same species. From the DOSY spectrum, it is difficult to judge if all signals have the same D because the spectrum has a very wide range. The spectrum should be shown in extended form (on the D-values axis), and the main text should contain a clear conclusion that all the signals belong to the same species.

3. A model of the encapsulation complex should be given, especially since the schematic representation in the figures does not seem to represent the postulated structure. The text says that the protons from the guests' phenyl groups are not encapsulated, whereas the schematic representation of the complex shows otherwise.

4. The paper should be checked for grammatical consistency. A large number of sentences have problems with logic and matching between the subject and the predicate. Just for example "Encapsulation studies using tertiary ammonium cations, which are typical guests for the hexameric capsule, have been studied but mostly in solution and the solid-state". I have found many such examples along the text, which I do not list here because it is not the role of the reviewer to do a grammar check.

5. Technical issues. Corrections of the chemical nomenclature are required (e.g. up-field should be upfield). Figures in the supplementary should have their unique names, e.g. Fig. S1. Fancy words such as 'de-excitation transitions' should be removed and replaced with commonly used terms such as emission, in this case.

Reviewer #2 (Remarks to the Author):

The authors have carefully addressed my concerns regarding the structural characterization of their complexes. I would like to see this nice paper published on NC as it is now.

Response to referees

REVIEWER REPORT(S):

Reviewer 1: Comments to the Author

Overall: The authors have made a great effort to explain the structural aspects of the formation of the symmetric Cl encapsulation complex of resorcin[4]arene with the luminescent Ir complex. In the revised version of the manuscript, many crucial issues have been clarified. The authors admitted their initial error in assigning the structure as pentameric and corrected their assignment to the hexameric capsule (based on the analysis of 2D NMR spectra). This seems to be a large structural difference; however, in fact, it has a minor effect on the general conclusion, which still holds. The authors addressed the questions and undoubtedly the quality of the article improved considerably. However, the paper still needs some substantial corrections and additional interpretations (major revision).

Answer: We appreciate the reviewer's consideration. We would carefully address again the reviewer's comments and suggestions as follows.

Comment 1: On the basis of the provided data for the complexation of Ir complexes containing various counterions, I think that the role of anions is still underestimated and not interpreted properly in this paper. Most likely, the anions participate in self-assembly, which is consistent with all the NMR data in this paper and is also related to recent findings (JACS 2022, 5350). For the resorcin[4]arene capsule that encapsulates the Ir-complex-Cl, the substantial downfield shift of OH groups is observed (compared to the free capsule and complexes containing other counterions). Therefore, anions may form hydrogen bonds with OH groups, rather than be co-encapsulated. This conclusion is in agreement with smaller changes observed during encapsulation of Ir-complex-NO₃⁻ (the NO₃⁻ anion, due to charge distribution, forms much weaker interactions with OH groups). The interpretation is also in agreement with the lack of encapsulation of Ir-complex-BF₄ or Ir-complex-PF₆, which contain anions that do not interact with the OH groups. The anions may be crucial for symmetry lowering, therefore I insist on a better discussion of the role of the anions in the paper. It is particularly relevant because changes in chiroptical properties observed for encapsulation also seem to depend on counterions.

Answer: As the reviewer pointed out, we cannot exclude the possibility that the Cl anion is trapped by OH groups to form an anion-sealed resorcin[4]arene hexameric capsule, although both resorcin[4]arene and pyrogallol[4]arene hexamers showed complete disassembly in the presence of 4 equiv of the ammonium salts in CHCl₃ (JACS 2022, 5350 and ref 33 in the main text). In fact, the crystal structure in our study using resorcin[4]arene and dicationic Ru complex clearly showed the Cl anion stabilized by the hydrogen-bonds with OH groups of the hosts. In the paragraph regarding the effect of counter anions in main text, we added a new discussion on the formation of the anion-sealed capsule. As the reviewer suggested, the ¹H NMR chemical shifts of OH groups might be effective indicators for the position of the Cl anion in the assembly. However, it is still indirect evidence and the chemical shifts should be carefully discussed because it is very sensitive to the relative orientation of the guest Ir complex. The Ir complex contains three aromatic organic ligands, which induce upfield or downfield shifts of signals of the OH groups by shielding effects from the aromatic ligands and CH...OH hydrogen bonds with the organic ligands. It should be noted that the solution structure of the host-guest complex still involves uncertainty. Thus, we would also keep another possibility to form an ion-pair within the neutral hexameric capsule, because molecular recognition within the resorcin[4]arene hexameric capsule had been realized to form ion-pairs of the guest salts (ref 33 in the main text). Although direct evidence of the position of Cl anions in solution is difficult to be determined by spectroscopic analyses, these results definitely show that the cooperative effect among the Cl anion, the guest cation, and the hydrogen-bonded hosts play a crucial role to build

up this remarkable supramolecular system. We sincerely thank the reviewers for the deep considerations about the role of the anion.

Comment 2: When symmetry lowering is discussed, it is crucial to know if all signals belong to the same species. From the DOSY spectrum, it is difficult to judge if all signals have the same D because the spectrum has a very wide range. The spectrum should be shown in extended form (on the D-values axis), and the main text should contain a clear conclusion that all the signals belong to the same species.

Answer: According to reviewer's suggestion, we freshly reprocessed the spectra and obtained the diffusion coefficient ($D = 3.6 \pm 0.4 \times 10^{-10} \text{ m}^2 \text{ s}^{-1}$) of the host-guest complex with a smaller error, which shows the formation of a single species more clearly. The revised ^1H DOSY spectrum was added in the main text as Figure 2f and the full spectrum was replaced with Supplementary Figure 4. In addition, we now clearly describe that the ^1H DOSY spectrum shows a single diffusion coefficient in the main text, implying that all the signals belong to the same species. We appreciate the reviewer's suggestion.

Comment 3: A model of the encapsulation complex should be given, especially since the schematic representation in the figures does not seem to represent the postulated structure. The text says that the protons from the guests' phenyl groups are not encapsulated, whereas the schematic representation of the complex shows otherwise.

Answer: As the reviewer suggested, we manually constructed a model structure of the host-guest complex and added it in the SI as Supplementary Fig. 14. It should be emphasized that the solution structure of the assembly is still mysterious to some extent, for example, concerning the position of the anion and the hydrogen-bonding pattern in the assembly. Thus, the proposed model structure roughly visualizes the relative orientation between the Ir complex cation and six resorcin[4]arene units in solution, based on the 2D NMR studies. In addition, we modified the schematic structure of the host-guest complex in the figures, as the reviewer pointed out. The new chemical representation in the figures (Figs. 2 and 3 as well as the figures in Supplementary Information) enhanced understandability of the solution structure that we speculated. We sincerely thank the reviewer for the kind comments.

Comment 4: The paper should be checked for grammatical consistency. A large number of sentences have problems with logic and matching between the subject and the predicate. Just for example "Encapsulation studies using tertiary ammonium cations, which are typical guests for the hexameric capsule, have been studied but mostly in solution and the solid-state". I have found many such examples along the text, which I do not list here because it is not the role of the reviewer to do a grammar check.

Answer: We carefully checked the manuscript and modified to improve readability of this paper. We appreciate the reviewer's comment.

Comment 5: Technical issues. Corrections of the chemical nomenclature are required (e.g. up-field should be upfield). Figures in the supplementary should have their unique names, e.g. Fig. S1. Fancy words such as 'de-excitation transitions' should be removed and replaced with commonly used terms such as emission, in this case.

Answer: As the reviewer pointed out, we carefully checked the scientific words and corrected them accordingly. On the other hands, names of the supplementary figures in the main text were unchanged, following Journal's

author guideline in Nature Communications (<https://www.nature.com/ncomms/submit/how-to-submit>). We thank the reviewer for the careful considerations.

Reviewer 2: Comments to the Author

Overall: The authors have carefully addressed my concerns regarding the structural characterization of their complexes. I would like to see this nice paper published on NC as it is now.

Answer: We really thank the reviewer for the warmful comment.

REVIEWER COMMENTS

Reviewer #1 (Remarks to the Author):

The authors have addressed my concerns regarding the participation of anions – some comments have been added. They also introduced other corrections (the model in SI and the corrected Figure). I recommend the publication of this paper in NC as it is now.

Response to referees

REVIEWER REPORT(S):

Reviewer 1: Comments to the Author

Overall: The authors have addressed my concerns regarding the participation of anions – some comments have been added. They also introduced other corrections (the model in SI and the corrected Figure). I recommend the publication of this paper in NC as it is now.

Answer: We really thank the reviewer for the valuable comments.